# Protein-directed ribosomal frameshifting temporally regulates gene expression

Sawsan Napthine[1,*], Roger Ling[1,*], Leanne K. Finch[1,*], Joshua D. Jones[1,*], Susanne Bell[1], Ian Brierley[1] & Andrew E. Firth[1]

Programmed −1 ribosomal frameshifting is a mechanism of gene expression, whereby specific signals within messenger RNAs direct a proportion of translating ribosomes to shift −1 nt and continue translating in the new reading frame. Such frameshifting normally occurs at a set ratio and is utilized in the expression of many viral genes and a number of cellular genes. An open question is whether proteins might function as *trans*-acting switches to turn frameshifting on or off in response to cellular conditions. Here we show that frameshifting in a model RNA virus, encephalomyocarditis virus, is *trans*-activated by viral protein 2A. As a result, the frameshifting efficiency increases from 0 to 70% (one of the highest known in a mammalian system) over the course of infection, temporally regulating the expression levels of the viral structural and enzymatic proteins.

---

[1] Division of Virology, Department of Pathology, University of Cambridge, Cambridge CB2 1QP, UK. * These authors contributed equally to this work. Correspondence and requests for materials should be addressed to I.B. (email: ib103@cam.ac.uk) or to A.E.F. (email: aef24@cam.ac.uk).

Programmed −1 ribosomal frameshifting (−1 PRF) is utilized in the expression of many viral genes and a number of cellular genes[1–3]. It may also play a widespread role in fine-tuning gene expression via the stimulation of nonsense-mediated mRNA decay pathways[4]. Sites of −1 PRF generally comprise a 'slippery' sequence (at which the change in reading frame occurs) and a 3′-adjacent stimulatory mRNA structure. In eukaryotes, the slippery sequence fits a consensus heptanucleotide motif X_XXY_YYZ, where XXX is any three identical nucleotides (although certain exceptions occur, such as GGU); YYY represents AAA or UUU; Z represents A, C or U; and underscores separate zero-frame codons[5]. In the tandem slippage model, the ribosomal P-site tRNA anticodon re-pairs from XXY to XXX, and the A-site anticodon re-pairs from YYZ to YYY, thus allowing perfect re-pairing except at wobble positions[6]. The efficiency of PRF is influenced by the identity of the slippery site nucleotides but is normally less than 1% in the absence of additional stimulatory elements. Thus, most known instances of eukaryotic −1 PRF are stimulated (typically to a level of 5–45%, depending on the particular case) by the presence of a 3′ stable RNA structure, such as a pseudoknot or stem–loop, separated from the slippery heptanucleotide by a 'spacer' region of 5–9 nt. Structures of this type are thought to be located at the mRNA unwinding site of the ribosome entrance channel when their stimulatory effect is exerted[7]. How the stimulatory RNAs function to promote −1 PRF is still uncertain, but accumulating evidence from prokaryotic counterparts indicates that the RNA structure impedes back rotation of the ribosomal small subunit, trapping the ribosome in a rotated or hyper-rotated state[8,9]. This stalled state can be resolved either via spontaneous unwinding of the structure or via a −1 PRF, which, by repositioning the structure within the mRNA entrance channel, allows for more efficient unwinding by the ribosome[9].

Like other members of the family Picornaviridae, encephalomyocarditis virus (EMCV; genus Cardiovirus) has a single-stranded RNA genome of positive polarity, which also serves as an mRNA. The genome is polyadenylated but lacks a 5′ cap and translation initiation is mediated by an internal ribosome entry site (IRES) within the lengthy 5′ UTR[10]. There is a single long open reading frame whose translation produces a polyprotein that is proteolytically cleaved, mainly by the virus-encoded 3C protease, to produce the structural and enzymatic proteins (Fig. 1a). For more than 50 years, EMCV has been used as a model system for investigating molecular virology, virus–host interactions and eukaryotic protein synthesis[11–13]. Studies of EMCV, besides poliovirus and foot-and-mouth disease virus, also led to the discovery of internal ribosome entry and StopGo co-translational separation, with the former now thought to be relevant for some key cellular genes and the latter an important tool in biotechnology[10,14–16]. Recently, we uncovered a further unusual aspect of EMCV translation, namely that a previously undetected −1 PRF site in an internal region of the polyprotein ORF directs a proportion of ribosomes into a short overlapping ORF resulting in the production of a 'trans-frame' protein, 2B⋆ (Fig. 1a)[17]. Our previous work suggested that the PRF mechanism in EMCV, and its relative Theiler's murine encephalomyelitis virus (TMEV), is atypical due to the apparent absence of an appropriately positioned stimulatory RNA structure, and a failure to reconstiute PRF outside of the context of virus infection[17,18].

Here, we describe the discovery of a protein trans-activator of EMCV frameshifting, viral protein 2A. We show that 2A binds to an RNA stem–loop beginning 14 nt downstream of the slippery sequence, forming an RNA:protein complex that induces highly efficient PRF. Through ribosome profiling, we investigate protein synthesis in the natural context of EMCV infection and find, remarkably, that frameshifting is temporally regulated, increasing from negligible levels at early time-points of infection to an efficiency of ∼70% at late time-points. Thus, frameshifting serves as a control mechanism to modulate the relative levels of viral structural and non-structural protein synthesis during the replicative cycle. At early time points, ribosome progression to the replicase coding sequences at the 3′ end of the genome is uninterrupted, but, as the replication cycle proceeds, frameshifting progressively diverts ribosomes from the polyprotein ORF to the overlapping 2B⋆ ORF, downregulating replicase translation. These experiments identify and elucidate a mechanism whereby the efficiency of PRF can be temporally regulated, in contrast to the fixed efficiency levels of canonical −1 PRF.

## Results

**Ribosome profiling shows a temporal shift in PRF efficiency.** To investigate PRF efficiency in the context of virus infection, we used ribosome profiling (Ribo-Seq) to directly monitor the density of ribosomes upstream and downstream of the frameshift site. Ribosome profiling is a recently developed technique that enables global footprinting of translating ribosomes in cells via high-throughput sequencing of ribosome-protected fragments (RPFs) (Fig. 1b)[19,20]. We infected murine L929 cells with wild-type (WT) or shift site mutant (SS) viruses and harvested cells at 2, 4, 6 and 8 h post infection (p.i.). In the SS mutant, the WT shift site G_GUU_UUU is mutated to A_GUG_UUU (Fig. 1c) to inhibit PRF[17]. Ribo-Seq libraries were prepared from each sample, deep sequenced, and the resulting reads mapped to host and viral genomes. Ribo-Seq quality was assessed as described previously (Supplementary Fig. 1)[21].

Figure 1d shows the distributions of RPFs on the WT and SS virus genomes at 4 and 8 h p.i. (see Supplementary Fig. 2a for 2 and 6 h p.i.). For WT virus, a substantial decrease in mean ribosome density was observed to occur after the 2B⋆ ORF at 6 and 8 h p.i., indicating that the majority of ribosomes translating the viral genome frameshift into the 2B⋆ ORF and terminate at the 2B⋆ stop codon. For the SS mutant, however, there is no significant drop in RPF density at the 2B⋆ stop codon, consistent with inhibited frameshifting in this mutant. PRF efficiencies were estimated from the WT ratio of downstream to upstream RPF densities, normalized by the SS mutant to control for variation in translational speed, technical biases and ribosome pausing between different genomic locations. PRF efficiencies were not obviously different from zero at 2 and 4 h p.i., but at 6 and 8 h p.i. were remarkably high, at 64% and 69%, respectively (Fig. 1e and Supplementary Fig. 2b). Consistent with the absence of measurable PRF at 2 and 4 h p.i., a noticeable ribosomal pause that was observed in the 2B⋆ region at 6 and 8 h p.i. for WT virus, is absent at 2 and 4 h p.i. and in the SS mutant (Fig. 1d and Supplementary Fig. 2a). The effector of this pause is uncertain though, in common with many other ribosomal pauses, it may be nascent peptide mediated. Extraordinarily, from 6 h p.i. RPFs accumulated to high levels on the mutated shift site (Fig. 1d and Supplementary Fig. 2a). In contrast, only a modest accumulation was apparent on the WT shift site. The position of RPFs in this region indicate that the pause initiates when the UUU of the shift site is in the ribosomal A-site and, in these samples, continues for ∼3 codons thereafter (Supplementary Fig. 2c,d), although such broadening of the pause is likely an artefact resulting from ribosome run-on during sample preparation[22]. The total excess accumulation of RPFs over these four codons at 6 and 8 h p.i. relative to 4 h p.i. is ∼5-fold greater in SS than in WT.

Ribosome profiling of eukaryotic systems typically has the characteristic that mappings of the 5′ end positions of RPFs to coding sequences reflect the triplet periodicity (herein referred to

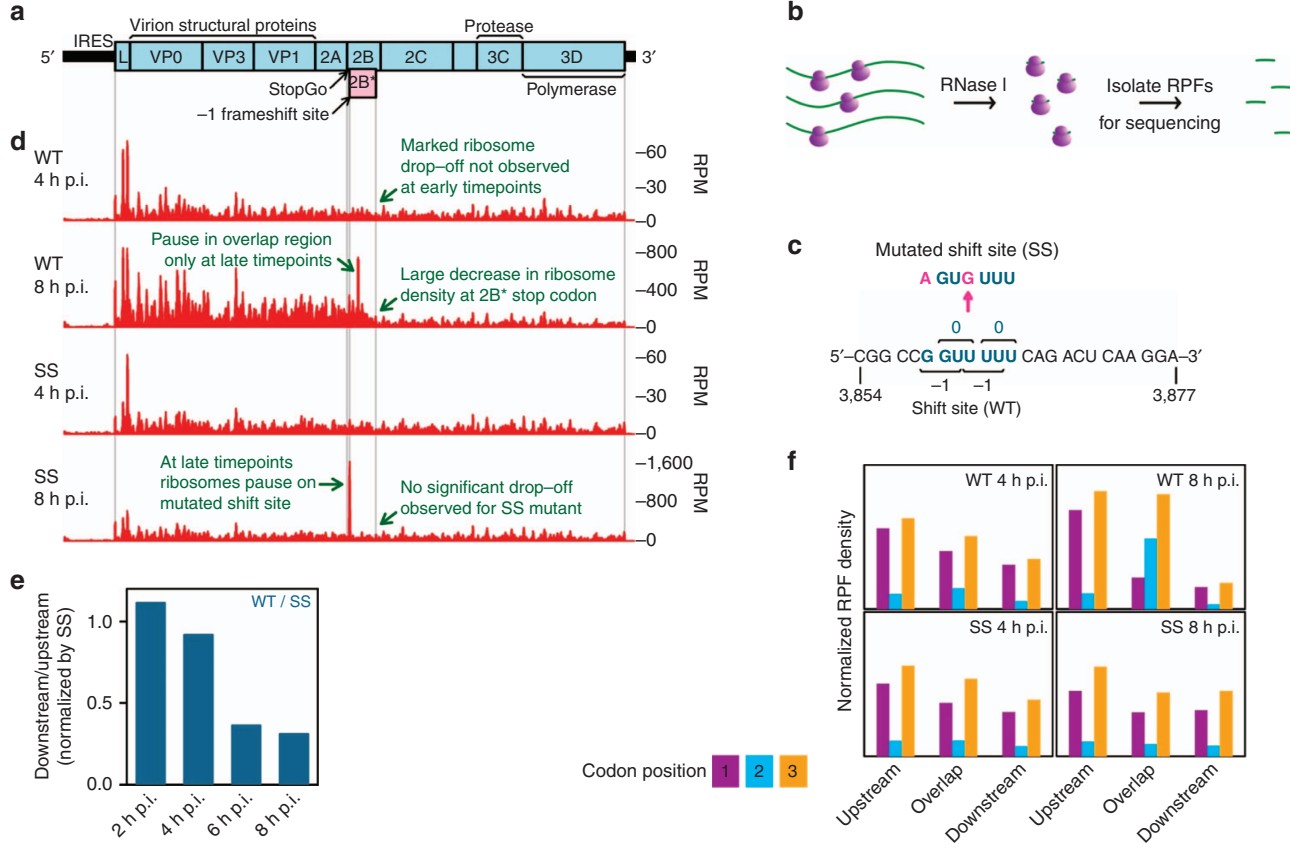

**Figure 1 | Frameshifting increases dramatically as infection progresses.** (**a**) Map of the ~7700-nt EMCV genome. The 5′ and 3′ UTRs are indicated in black and the polyprotein ORF is indicated in pale blue with subdivisions showing mature cleavage products. The overlapping 2B* ORF is indicated in pale pink. (**b**) Schematic of the ribosome profiling strategy. Each translating ribosome protects ~30 nt of mRNA. Cells are lysed, treated with RNase I to degrade unprotected mRNA, ribosomes are harvested and the ribosome protected fragments (RPFs) extracted and subjected to high-throughput sequencing. (**c**) Mutations introduced to prevent PRF. (**d**) Ribo-Seq RPF densities in reads per million mapped reads (RPM) on WT and SS virus genomes at 4 and 8 h p.i. (**e**) Ratio of downstream to upstream RPF densities in WT virus divided by the corresponding ratio in SS mutant virus. (**f**) Phasing of RPFs mapping upstream of 2B*, within the 2B/2B* overlap region, and downstream of 2B*.

as 'phasing') of genetic decoding. Figure 1f and Supplementary Fig. 2e show the polyprotein-frame codon positions to which the 5′ ends of RPFs map for regions upstream of 2B*, within 2B*, and downstream of 2B*. For the upstream and downstream regions, RPF 5′ ends map mainly to the first and third positions of polyprotein-frame codons. In contrast, within the 2B* region, RPF 5′ ends for WT virus at 6 and 8 h p.i. map mainly to the third and second positions of polyprotein-frame codons, consistent with a mixture of ribosomes translating the two overlapping reading frames in this region. For the SS mutant, there is no change in the phasing of RPF 5′ end positions in the overlap region (Fig. 1f), consistent with inhibited PRF in this mutant.

**An RNA stem–loop is essential for PRF and ribosome pausing.** Previously, a downstream stem–loop structure, separated from the frameshift site by a 13-nt 'spacer', was identified bioinformatically[17]. This positioning is inconsistent with canonical mRNA structure stimulators of − 1 PRF, which are separated from the shift site by just 5–9 nt. The predicted stem–loop is conserved in different isolates of EMCV and in TMEV, with compensatory substitutions (i.e. paired substitutions that preserve the predicted base-pairings) further supporting its biological relevance (Supplementary Fig. 3a). RNA structure probing also supported the stem–loop (Supplementary Fig. 3b,c). To

assess a role for the stem–loop in PRF and/or ribosome pausing, we generated a stem–loop mutant virus, WT-SL, and a corresponding shift site mutant, SS-SL, and performed ribosome profiling at 8 h p.i. (Fig. 2a,b; see Supplementary Fig. 4 for Ribo-Seq quality assessment and Supplementary Fig. 5 for corresponding RNA-Seq data). Pausing at the mutated shift site (SS) was absent in the stem–loop mutant (SS-SL), showing that the stem–loop is required for pausing. Assuming an average translation speed[20] of 0.18 s codon[−1], the excess accumulation of RPFs in SS (Fig. 2c) equates to a ribosomal pause of ∼20 s (albeit with certain caveats; see Methods). For SS, WT-SL and SS-SL viruses, there was no marked change in RPF phasing in the overlap region (Fig. 2d), consistent with inhibition of PRF. The PRF efficiency was measured as 70% for WT virus, but negligible for the SS, WT-SL and SS-SL mutants (Fig. 2e). Thus the stem–loop is essential for efficient PRF.

The EMCV frameshift signal is located just downstream of the junction between the 2A- and 2B-encoding regions of the polyprotein ORF (Figs 1a and 2a). Separation between 2A and 2B occurs co-translationally via a mechanism known as 'StopGo' or 'Stop-Carry On' that depends critically on the amino acid motif D(V/I)ExNPGP (where the last proline is the first amino acid of 2B)[15,16]. To assess whether StopGo affects PRF, we applied ribosome profiling to viruses LV-WT and LV-SS-SL in which StopGo was inhibited by mutating the NPGP sequence to NPLV (Fig. 2a,b)[23]. For LV-WT, RPF phasing in the overlap region was

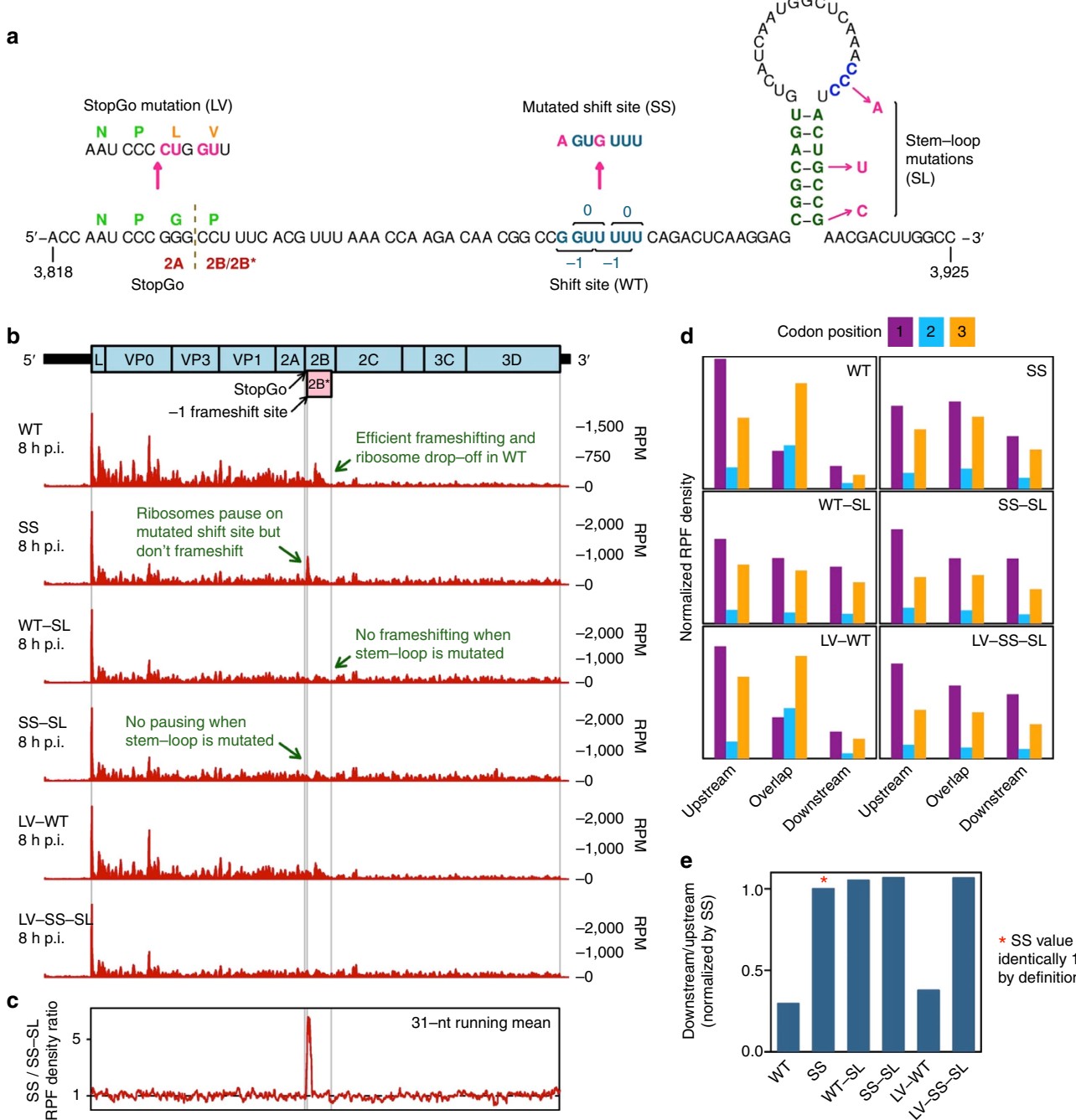

**Figure 2 | An RNA stem–loop is required for frameshifting.** (**a**) Mutations introduced to prevent PRF (SS) or StopGo-mediated co-translational separation at the C-terminus of 2A (LV) or to destabilize the predicted RNA stem–loop (SL). (**b**) Ribo-Seq RPF densities in reads per million mapped reads (RPM) on WT and mutant virus genomes at 8 h p.i. WT and SS samples are biological repeats of the 8 h p.i. samples in Fig. 1. (**c**) Ratio of SS and SS-SL RPF distributions after smoothing with a 31-nt running-mean filter. (**d**) Phasing of RPFs mapping upstream of 2B*, within the 2B/2B* overlap region, and downstream of 2B*. (**e**) Ratio of downstream to upstream RPF densities in the different viruses divided by the corresponding ratio in SS mutant virus.

not greatly different from WT (Fig. 2d), and the calculated PRF efficiency (62%) was only slightly lower than WT (70%) (Fig. 2e). The modest difference may be due to slower replication in the mutant leading to a lag in achieving maximum PRF efficiency, or may indicate a more direct (albeit modest) stimulatory effect of StopGo on PRF, likely related to proper processing of the C terminus of 2A (see below). On the other hand, for the mutant LV-SS-SL these features were similar to the SS-SL mutant. Thus we found no evidence for ribosome drop-off induced by StopGo alone.

**Virus protein 2A binds the RNA stem–loop.** Having observed that PRF in EMCV increases from 0 to 70% over the course of infection and depends critically on an RNA stem–loop structure positioned unusually far 3′ of the frameshift site, we reasoned that some virus-induced *trans*-acting protein might interact with the stem–loop to promote PRF. To test this, we used the RiboTrap system in which RNA transcripts are labelled with 5-bromo-uridine to allow immunopurification of bound complexes. A short RNA transcript containing the EMCV stem–loop was found to bind an ∼16 kDa protein from

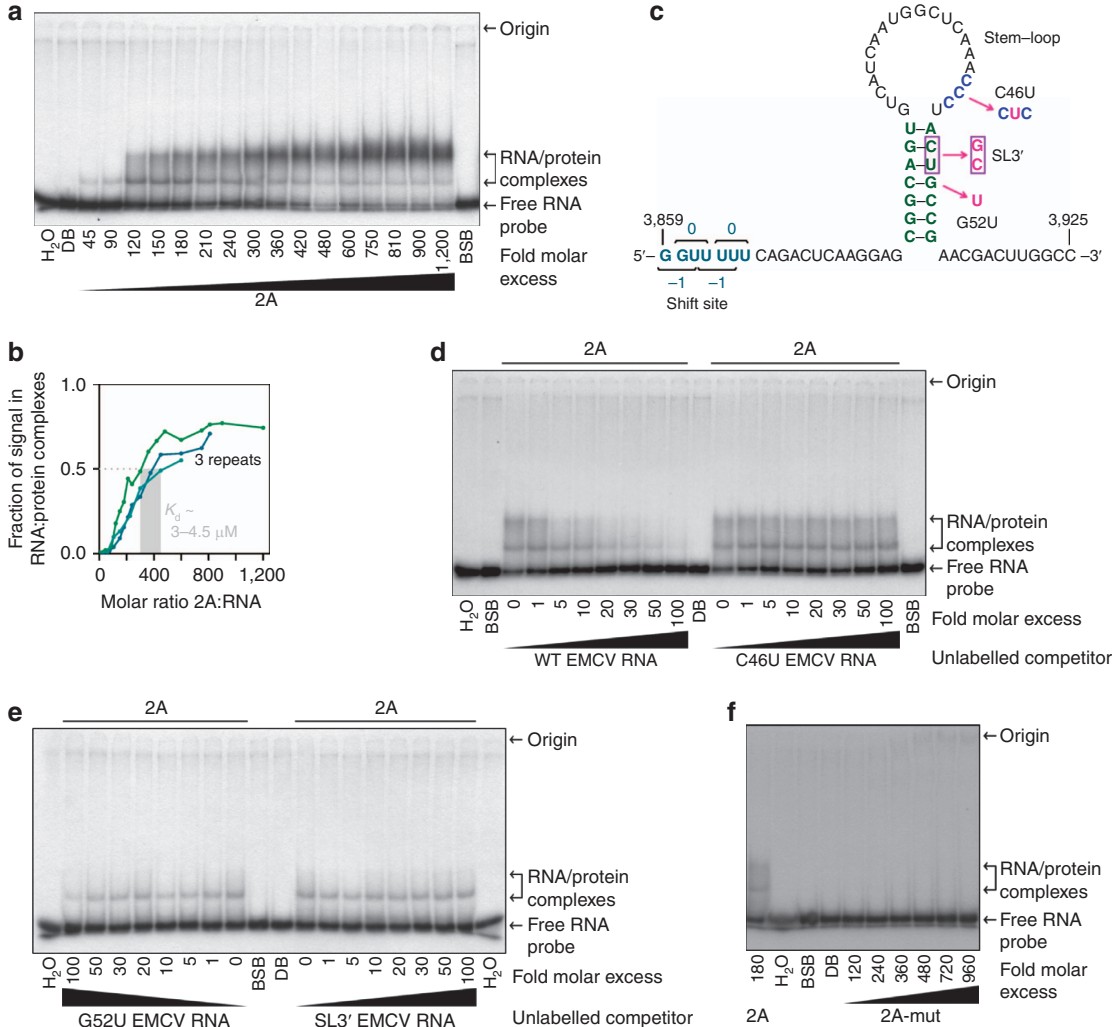

**Figure 3 | Viral protein 2A binds the stem–loop.** (**a**) EMSA analysis of binding of 2A to a 64-nt [32]P-labelled RNA containing the EMCV PRF signal. Numbers below lanes show fold molar excess of 2A with respect to RNA (10 nM). In lanes BSB, DB and H$_2$O, RNA was incubated alone with band-shift buffer, protein dilution buffer or water, respectively. (**b**) Phosphorimager quantification of RNA in RNA:protein complexes for (**a**) and two further repeats. (**c**) Mutations introduced into the stem–loop for competition assays. (**d,e**) Unlabelled WT or C46U, G52U or SL3′ mutant competitor RNA was incubated with WT [32]P-labelled RNA and 2A (1.8 μM), and analysed by EMSA. Numbers below lanes show fold molar excess of competitor RNA with respect to [32]P-labelled WT RNA (10 nM). (**f**) WT [32]P-labelled RNA was incubated with increasing amounts of 2A-mut, and analysed by EMSA. WT 2A was used as a control.

EMCV-infected cell lysates that was not observed with mock-infected cell lysates or with a control RNA containing a scrambled version of the EMCV stem–loop sequence (Supplementary Fig. 6a). Mass spectrometric analysis of this product revealed it to be the viral protein 2A (Fig. 1a and Supplementary Fig. 6b).

To confirm this interaction, we expressed and purified recombinant 2A protein and performed electrophoretic mobility shift assays (EMSAs) with a 64-nt [32]P-labelled RNA containing the EMCV frameshift site and stem–loop. Two main RNA:protein complexes were observed, with the more slowly migrating species accumulating as the amount of 2A was increased (Fig. 3a), confirming that 2A does indeed bind the EMCV PRF signal. We estimated the $K_d$ of the interaction to be ~3–4.5 μM (Fig. 3b). A CCC triplet in the RNA loop is highly conserved between different isolates of EMCV and TMEV (Supplementary Fig. 3a), suggesting a potential role in 2A binding. To investigate this, we mutated the middle C to U (C46U; Fig. 3c) and tested whether cold WT RNA or cold mutant

RNA could compete with the [32]P-labelled WT RNA for 2A binding. Whereas the WT RNA was able to compete well, greatly diminishing radiolabelled RNA:protein complexes at increasing molar excess, the C46U mutant was unable to compete, indicating that this single-nucleotide mutation had prevented 2A binding (Fig. 3d). Similarly, RNAs with stem-disrupting mutations (G52U and SL3′; Fig. 3c) were also unable to compete with WT RNA (Fig. 3e), indicating that the presence of at least part of the stem–loop duplex is also important for 2A binding (and not just for proper positioning of the RNA:protein complex for PRF stimulation). A direct interaction between 2A and RNA would likely involve a cluster of positively charged residues on 2A. By inspecting an alignment of cardiovirus 2A sequences we identified a conserved linear basic cluster (Supplementary Fig. 6c). We generated a recombinant mutant 2A (hereafter 2A-mut) by changing R95 and R97 to alanines (Supplementary Fig. 6d,e). EMSA analysis confirmed that 2A-mut was unable to bind to the EMCV PRF signal (Fig. 3f). Further, when introduced into the virus genome, the

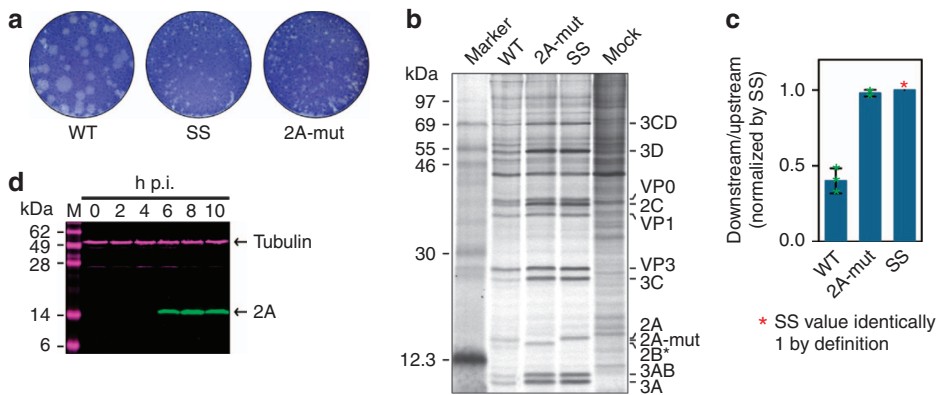

**Figure 4 | Mutating 2A knocks out frameshifting.** (**a**) Plaque morphology of WT, SS and 2A-mut viruses on BHK-21 cells (see also ref. 25). (**b**) Metabolic labelling of BHK-21 cells mock-infected or infected with WT, SS or 2A-mut viruses. Positions of EMCV proteins are indicated. (**c**) Mean ratio of measureable polyprotein products encoded downstream of the frameshift site to products encoded upstream of the frameshift site, normalized by the SS mutant. Bars show means ± s.d. of three biological replicates (green crosses). (**d**) Time course of 2A expression during WT virus infection assessed by immunoblotting using antibodies to 2A (green) and tubulin (magenta; loading control).

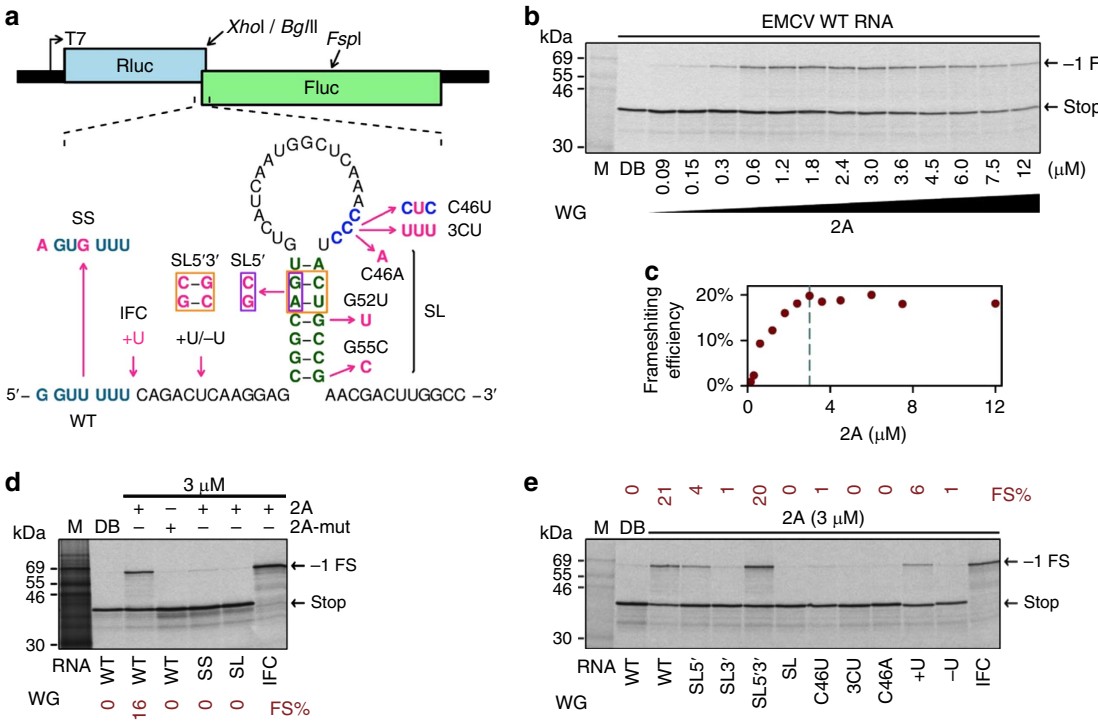

**Figure 5 | Analysis of the EMCV frameshift signal.** (**a**) Mutations introduced into the EMCV PRF signal in reporter plasmid pDluc. (**b**–**e**) RNAs derived from FspI-cut plasmid were translated in WG extract in the presence of (**b**) increasing concentrations of 2A or (**d**,**e**) 3 μM 2A or 2A-mut, or 2A dialysis buffer (DB). Products generated by ribosomes that do not frameshift (stop) or that enter the −1 reading frame (−1 FS) are indicated. M and IFC indicate markers and the in-frame control, respectively. Panel **c** shows the PRF efficiencies for **b**; the dashed line indicates the 2A concentration used in (**d**,**e**).

R95 and R97 substitutions resulted in a protein expression pattern similar to that of SS mutant virus, indicating that the 2A mutations had inhibited PRF (Fig. 4a–c). Consistent with the increase in frameshifting efficiency, the amount of 2A in cells infected with WT virus increases dramatically between 4 and 6 h p.i. (Fig. 4d).

**Virus protein 2A *trans*-activates ribosomal frameshifting.** Next we asked whether the 2A protein is sufficient to stimulate PRF in the absence of viral infection. To test this we titrated recombinant

2A into a wheat germ (WG) *in vitro* translation system programmed with a reporter mRNA containing the EMCV PRF signal (Fig. 5a). Increasing amounts of 2A led to a modest general inhibition of cap-dependent translation, as reported previously[24,25]. When the reporter was translated in the absence of 2A, only the non-frameshift product was observed (Fig. 5b, lane DB). However, in the presence of 2A, efficient PRF was observed, to a level of ~20% with increasing amounts of 2A (Fig. 5b,c). In control translations, PRF was abolished upon mutating the frameshift site (SS), or the stem–loop and CCC triplet (SL), or upon replacing 2A with 2A-mut (Fig. 5d).

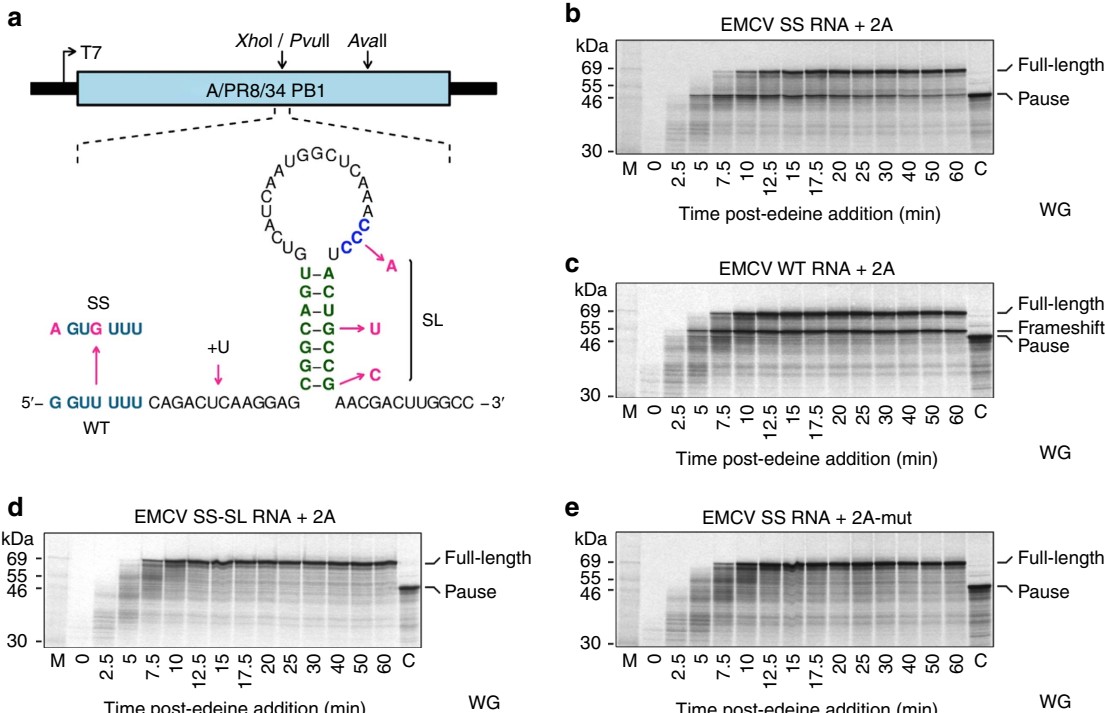

**Figure 6 | Ribosomal pausing at the EMCV frameshift signal.** (**a**) Mutations introduced into the EMCV PRF signal in reporter plasmid pPS0. (**b–e**) RNAs derived from *Ava*II-cut plasmids were translated in WG extract, after 5 min further initiation was halted by the addition of edeine, and aliquots were removed at various times and analysed by SDS–PAGE. Lanes M and C show markers and the expected size of the ribosomal pause product, respectively. Translations were supplemented with 1 μM 2A (**b–d**) or 2A-mut (**e**). As well as the full-length product and the transient pausing product, a frameshift product is produced for WT RNA only (**c**).

Changing the loop-region CCC triplet to UUU (3CU), CUC (C46U) or CAC (C46A) also inhibited PRF (Fig. 5e). The SL5′ and SL3′ mutations of Figs 3c and 5a substantially reduced PRF (to 4% and 1%, respectively) whereas the SL5′3′ mutations (designed to restore the stem–loop structure but with altered base pairings) restored PRF to WT levels (Fig. 5e). Further, increasing (+U) or decreasing (−U) the spacer length by 1 nt reduced PRF to 6 and 1%, respectively (Fig. 5e). Similar results were observed in the rabbit reticulocyte lysate (RRL) *in vitro* translation system (Supplementary Fig. 7).

The *in vitro* translation systems were also used to further investigate ribosome pausing at the frameshift site (Fig. 6 and Supplementary Fig. 8). The extent of pausing was assessed by comparing the levels of a translational intermediate corresponding to pausing at the EMCV shift site with that of the full-length polypeptide produced during a time course in which translation was synchronized by the addition of edeine, a potent inhibitor of initiation, 5 min after the start of the reaction. Consistent with the ribosome profiling, we observed strong pausing at a mutated shift site (SS) in the presence of recombinant 2A (Fig. 6b and Supplementary Fig. 8a), a much less pronounced pause at the WT shift site (Fig. 6c and Supplementary Fig. 8b), and no pause when the stem–loop and CCC triplet were mutated (SS-SL and WT-SL; Fig. 6d and Supplementary Fig. 8c) or when SS or WT RNA was translated in the presence of 2A-mut (Fig. 6e and Supplementary Fig. 8d). Thus, both the stem–loop and 2A protein are required for ribosome pausing. When the spacer length was increased by 1 nt (SS + U and WT + U) pausing was observed in the presence of 2A but was much less pronounced than for SS RNA (Supplementary Fig. 8e,f). The appearance of the pause product is transitory (albeit spread over several minutes for WT and many minutes for SS; Fig. 6b,c), consistent with its identity as a genuine intermediate rather than a dead-end product. There was no

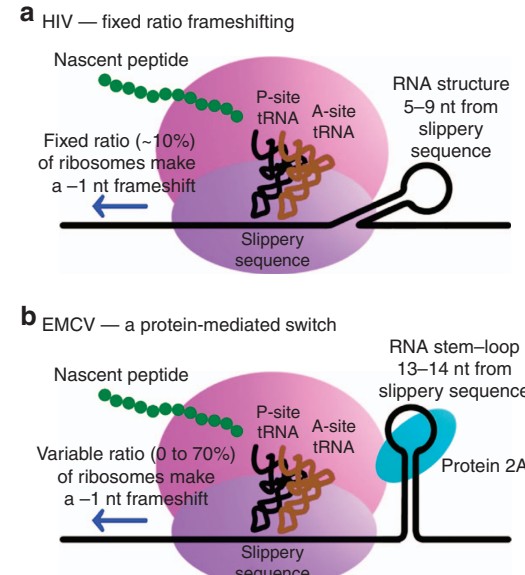

**Figure 7 | Comparison of −1 frameshifting stimulators.** (**a**) Nearly all known cases of −1 PRF are stimulated by an mRNA structure, such as a pseudoknot or stem–loop, separated from the slippery heptanucleotide shift site by a spacer sequence of 5–9 nt, leading to a constant ratio of frameshift to non-frameshift product. (**b**) In contrast, frameshifting in EMCV is stimulated by a protein:RNA complex positioned at the leading edge of the ribosome when the decoding centre is on the shift site. Increasing levels of viral protein 2A result in a 0 to 70% switch in frameshifting efficiency between early and late timepoints of infection.

evidence for significant ribosomal drop-off occurring at the WT shift site (Fig. 6c and Supplementary Fig. 8b, 60-min lane) though we could not rule out a small amount of drop-off occurring when the shift site was mutated due to the long half-life of the pause product for these constructs (Fig. 6b and Supplementary Fig. 8a).

## Discussion

We have shown that PRF in EMCV is extraordinarily efficient, and, in stark contrast to other cases of −1 PRF such as from HIV or SARS coronavirus, varies dramatically over the course of infection from 0 to ~70% and depends crucially on a *trans*-activating virus protein (Fig. 7). Early in infection the efficiency of PRF is negligible, allowing efficient translation of the 3′-encoded replication proteins. However, late in infection and presumably as a result of increasing concentrations of cytoplasmic 2A, PRF becomes highly efficient, resulting in >3-fold downregulation of replication protein synthesis. This provides an elegant and economic solution to facilitate maximal build up of replication capacity early in infection, while providing a translational bias that favours virion production at late timepoints, and raises the question of whether other 'single-polyprotein' viruses such as poliovirus and hepatitis C might also use non-canonical translational mechanisms to temporally regulate replication protein synthesis. These results provide a satisfying conclusion to observations by Paucha *et al.* who, 40 years ago, observed approximately twice as much capsid as noncapsid viral protein in EMCV-infected cells, specifically at late timepoints. Although they could not then provide a mechanism, these authors showed extraordinary insight—long before the EMCV genome was sequenced—by proposing a model in which, to quote, 'some viral-specified protein acts as a specific termination factor capable of causing premature termination of translation at a site located near the midpoint of the viral RNA'[26]. We can now confirm that this effect is mediated at the translational level, and depends on protein-stimulated PRF rather than ribosomal drop-off at StopGo or at other sites during polyprotein synthesis.

Together with −2 PRF for nsp2TF expression in members of the family Arteriviridae, where we recently demonstrated that PRF is critically dependent on the arteriviral protein nsp1β and host poly(C) binding proteins interacting with a 3′ C-rich motif separated from the frameshift site by a 10-nt spacer[27], this is one of only two known cases of protein-stimulated PRF. As 2A and nsp1β are viral proteins, they cannot be utilized for normal cellular gene expression. Nonetheless there may be other cases of protein-stimulated PRF and the two extant examples suggest that the protein binding sites can be positioned just outside the mRNA entrance channel at the onset of frameshifting. Cellular analogues would have been missed by previous bioinformatic searches for PRF sites that have relied on the presence of a predicted RNA structure beginning closer to a slippery heptanucleotide shift site[28].

An association between ribosomal pausing and RNA-structure-stimulated −1 PRF has been previously reported[29–31] though it has been unclear to what extent pausing is required for −1 PRF as opposed to being an unavoidable side effect. Of note, several previous pausing analyses used reporters with a mutated shift site. Similar to recent analyses of RNA structure-stimulated −1 PRF in a cell-free *Escherichia coli* system[9], we observed much more pronounced pausing when the shift site was mutated than for an intact shift site (Supplementary Fig. 2c,d in cells and Fig. 6b,c *in vitro*), suggesting that, by making a −1 nt shift, ribosomes are able to more efficiently remove 2A and unwind the stem–loop than if they remain in the original frame. Similar to that study,

but in contrast to recent single-molecule FRET work[8], the EMCV pause starts when the P- and A-sites are on the shift site, not upstream. The reason for the significantly lower PRF efficiency observed *in vitro* (maximum ~20%) compared to virus infection (maximum ~70%) remains unknown, but possibilities include incomplete activity of the recombinant 2A, effects of more distal RNA sequences, or differences in translational environment, such as salt, temperature, pH and ribosome loading. In the absence of authentically processed 2A, PRF in the StopGo (LV) mutants would be stimulated by 2A–2B (or cleaved versions thereof (ref. 17)) and/or 2A–2B* and this may explain the slightly reduced PRF level in LV-WT (Fig. 2e). The stem–loop:2A complex likely acts as an analogue of an RNA structure stimulator, except that it is regulatable. At this stage it is not known if it is simply an obstacle (e.g. to ribosome subunit rotation) or has some more specific interaction with the ribosome.

This report describes an example of a temporally regulated PRF 'switch'. A second example of viral protein-stimulated PRF has been noted recently in arteriviruses[27], and an miRNA-stimulated PRF signal has been discovered in the mRNA encoding the HIV-1 co-receptor CCR5 (ref. 4). The recent identification of these unrelated examples of *trans*-activated PRF signals indicates such signals may be more widespread in viruses and their hosts, and that regulated expression through the stimulation of −1 PRF will become an established paradigm of gene expression.

## Methods

**Cells, recombinant viruses and plasmids.** Cell lines were obtained from the European Collection of Authenticated Cell Cultures (ECACC) and tested for mycoplasma by PCR (e-Myco *plus* Mycoplasma PCR Detection Kit; iNtRON Biotechnology). In addition, the sequenced libraries (L929 cells) were queried for mycoplasma sequences. WT and mutant viruses are based on the EMCV subtype mengovirus cDNA, pMC0, developed by Ann Palmenberg (University of Wisconsin-Madison)[32]. The parental (WT) sequence used is similar to GenBank accession DQ294633.1, but a 55-nt poly(C) tract in the 5′ UTR is deleted and there are 13 single-nucleotide differences (A2669C, G3044C, C3371T, A4910C, G4991A, C5156T, G5289A, G5314C, G5315A, A5844C, G6266A, G6990A, A6992G; DQ294633.1 coordinates). Excepting the preceding sentence, nucleotide coordinates are given with respect to vMC0. WT, SS and LV-WT viruses were a kind gift from Gary Loughran (University College Cork)[17,23]. All constructs were prepared by standard PCR mutagenesis and recombinant DNA techniques and subcloned regions altered by mutagenesis were verified by DNA sequencing. All viruses were able to replicate in cell culture. The SS and SL mutations do not alter the polyprotein amino acid sequence.

***In vitro* frameshifting at the EMCV PRF signal.** For *in vitro* frameshifting assays, we cloned a 105-nt sequence containing the G_GUU_UUU shift site flanked by 12 nt upstream and 86 nt downstream, or mutant derivatives, into the dual luciferase plasmid pDluc at the *Xho*I/*Bgl*II sites[33]. The sequence was inserted between the *Renilla* and firefly luciferase genes so that firefly luciferase expression is dependent on −1 PRF. For ribosomal pausing analysis, the EMCV shift site flanked by 5 nt upstream and 93 nt downstream, or mutant derivatives, were cloned into pPS0 at the *Xho*I/*Pvu*II sites[29]. For the expression of recombinant 2A in *E. coli*, the 2A coding sequence was amplified from pMC0 and cloned into pGEX-6P-2 (GE Healthcare) at the *Bam*HI/*Xho*I sites. The expressed 2A, following removal of the glutathione-*S*-transferase moiety by PreScission Protease (a kind gift from Stephen Graham, University of Cambridge), has an additional 4 and 13 vector-derived residues at its N- and C-termini, respectively.

***In vitro* transcription and generation of recombinant virus.** RNA was transcribed using the Megascript T7 kit (Ambion) from *Bam*HI-linearized plasmids. Reactions were phenol/chloroform extracted, the RNA desalted by centrifugation through a NucAway Spin Column (Ambion) and concentrated by ethanol precipitation. Purified RNAs were used to transfect 35-mm dishes of L929 or BHK-21 cells using 1.2 μg RNA and 4 μl DMRIE-C reagent according to the manufacturer's instructions. Transfected cells were cultured in Dulbecco's modified Eagle's medium (DMEM) with high glucose and 1% fetal bovine serum (FBS) for 1–5 days depending upon how rapidly cytopathic effect developed. Cultures were subjected to three rounds of freeze–thawing, cell debris removed by centrifugation for 5 min at 4,000*g* and the supernatant stored in aliquots at −80 °C.

**Plaque assays.** BHK-21 cells at 90% confluence in six-well plates were infected with serial dilutions of virus stocks. Cells were washed with serum-free medium, overlaid with virus innoculum and incubated for 1 h at 37 °C. Innocula were removed and replaced with 1.5% low melting point agarose (Invitrogen) containing DMEM plus 2% FBS. After 40 h incubation at 37 °C, cells were fixed with formal saline and stained with 0.1% toluidine blue.

**Metabolic labelling and calculation of PRF efficiencies.** BHK-21 cells were infected at a multiplicity of infection (MOI) of ~10 in a volume of 150 μl in 24-well plates. After 1 h the inoculum was replaced with 1 ml DMEM containing 1% FBS. At 7 h p.i., cells were incubated for 1 h in methionine- and serum-free DMEM, and radiolabelled from 8 to 9 h p.i. with [35S] methionine at 100 μCi ml$^{-1}$ (~1,100 Ci mmol$^{-1}$) in methionine-free medium. Cells were scraped into the medium, pelleted at 13,000g for 1 min, washed twice by resuspension in 1 ml of ice-cold phosphate-buffered saline (PBS) and pelleted for 2 min at 13,000g. Cell pellets were lysed in 35 μl 4× SDS–PAGE sample buffer and boiled for 5 min before analysis by SDS–PAGE. Dried gels were exposed to X-ray films or to phosphorimager storage screens. Image analysis was carried out using Image-QuantTL 7.0, and the radioactivity in virus-specific products quantified.

The intensity for each WT virus product was measured, normalized by methionine content and then by the mean value for VP0, VP3 and VP1 to control for lane loading. Next, to factor out differences in protein turnover besides unquantified processing intermediates, for each biological replicate the WT values for VP0, VP3, VP1, 2C, 3A + 3AB, 3C + 3CD and 3D + 3CD were normalized by corresponding values for SS mutant virus. Then the normalized values for 2C, 3A + 3AB, 3C + 3CD and 3D + 3CD (i.e. products encoded downstream of the frameshift site) were averaged and divided by the average of the values for VP0, VP3 and VP1 (i.e. products encoded upstream of the frameshift site). This gives an estimate of the fraction of ribosomes that avoid a −1 PRF (Fig. 4c). One minus this value estimates the PRF efficiency.

**Ribosome profiling library preparation.** L929 cells in 60-mm dishes were infected at an MOI of ~10 in an initial volume of 1.5 ml DMEM with 1% FBS. After 1 h adsorption at 37 °C an additional 3.5 ml was added and incubation continued. At the appropriate time points, cells were treated with cycloheximide (Sigma-Aldrich; to 100 μg ml$^{-1}$; 2 min). Cells were rinsed with 5 ml of ice-cold PBS, the dishes submerged in a reservoir of liquid nitrogen for 10 s, transferred to dry ice and 400 μl of lysis buffer (20 mM Tris-HCl pH 7.5, 150 mM NaCl, 5 mM MgCl$_2$, 1 mM DTT, 1% Triton X-100, 100 μg ml$^{-1}$ cycloheximide and 25 U ml$^{-1}$ TURBO DNase (Life Technologies)) dripped onto the cells. The cells were scraped extensively to ensure lysis, collected and triturated with a 26-G needle 10 times. Lysates were clarified by centrifugation for 20 min at 13,000g at 4 °C, the supernatants recovered and stored in liquid nitrogen. Cell lysates were subjected to Ribo-Seq and RNA-Seq. The methodologies employed were based on the original protocols of Ingolia et al.[19,34], except library amplicons were constructed using a small RNA cloning strategy[35] adapted to Illumina smallRNA v2 to allow multiplexing[36]. RiboZero-based rRNA subtraction was used for RNA-Seq libraries (Epicentre, cat. no. RZH1046) while Ribo-Seq libraries were untreated. Amplicon libraries were sequenced using the Illumina HiSeq 2000 platform at the Beijing Genomics Institute (round 1) and the Illumina NextSeq platform at the Department of Biochemistry, University of Cambridge (round 2). Sequencing data have been deposited in ArrayExpress (http://www.ebi.ac.uk/arrayexpress) under the accession number E-MTAB-5206.

**Ribosome profiling data analysis.** Adaptor sequences were trimmed using the FASTX-Toolkit (Hannon lab) and reads shorter than 25 nt were discarded. Trimmed reads were first mapped to Mus musculus databases of rRNA, ncRNA (Ensembl, GRCm38.70, ncRNA) and mRNA (NCBI RefSeq mRNAs) using bowtie version 1 with seed length 23 and default parameters[37]. Remaining reads were mapped to the relevant virus genome (EMCV or mutants thereof). To ascertain that prior mapping to host sequences did not remove viral reads, we also tested mapping reads to the virus genome first, and found that the same set of reads were identified. Supplementary Table 1 shows the composition statistics for each library.

Host mRNA RiboSeq phasing distributions (Supplementary Figs 1c and 4c) were derived from reads mapping to the 'interior' regions of annotated coding ORFs; specifically, the 5′ end of the read had to map between the first nucleotide of the initiation codon and 30 nt 5′ of the last nucleotide of the termination codon, thus, in general, excluding RPFs of initiating or terminating ribosomes. Histograms of inferred approximate P-site positions (5′ end coordinate +12 nt offset) of host mRNA RPFs relative to initiation and termination codons (Supplementary Figs 1a and 4a) were derived from reads mapping to RefSeq mRNAs with annotated CDSs ≥450 nt in length and annotated 5′ and 3′ UTRs ≥60 nt in length. All figures are based on total numbers of mapped reads, rather than weighted sums for highly expressed mRNAs (cf. ref. 19), because virus-induced shut-off of host cell translation at late time points reduces the efficacy of the latter approach for our data. Read length distributions (Supplementary Figs 1b,4b and 5b) are based on total mapped reads (to positive-sense host mRNA or EMCV genome, as indicated) without restriction to annotated coding regions.

Plots showing RPFs mapped to the EMCV genome (Figs 1d and 2b and Supplementary Fig. 2a) show the positions of the 5′ ends of RPFs with a +12 nt offset to show the approximate P-site, smoothed with a 15-nt running-mean filter. For consistency, RNA-Seq plots (Supplementary Fig. 5) also show the positions of the 5′ ends of reads with a +12 nt offset. Read densities are displayed in reads per million reads mapped to the virus positive-sense or host messenger RNA (RPM).

To calculate PRF efficiencies from profiling data, Ribo-Seq read densities upstream and downstream of the PRF signal were determined. In this analysis, normalization by RNA-Seq densities was not carried out as the kinetics of RNA synthesis are not synchronous with the kinetics of RNA translation, and, especially late in infection, newly synthesized RNA may be destined for packaging rather than translation. Ribosome density along the virus genome may be affected by the kinetics of translation (as newly synthesized RNA enters the translation pool it takes 10–12 min for ribosomes to reach the 3′ regions[38]), variations in translation speed and ribosome pausing between different genomic regions, technical biases (nuclease, PCR, ligation)[39] and potentially ribosome 'drop-off' at other sites along the genome. The SS mutant was used as an appropriate control by which to normalize the WT RPF density for these factors. PRF efficiencies (Figs 1e and 2e) were calculated using RPFs with estimated P-sites mapping within the regions 180 nt after the polyprotein initiation codon to 180 nt upstream of the junction between the 2A and 2B coding regions (upstream density), and 180 nt downstream of the 2B* stop codon to 180 nt upstream of the polyprotein stop codon (downstream density). RPF counts were divided by region lengths to obtain RPF densities. To calculate the fraction of ribosomes that avoid a −1 PRF, the downstream mean density was divided by the upstream mean density, and this value for WT or mutant viruses was divided by the corresponding value for the SS mutant. One minus this value estimates the PRF efficiency.

For phasing histograms (Figs 1f and 2d and Supplementary Fig. 2e), RPF densities were calculated for each of the three phases (based on RPF 5′ end position within polyprotein-frame codons), for RPFs with estimated P-sites mapping within each of the three regions: from 30 nt after the polyprotein initiation codon to 30 nt upstream of the junction between the 2A and 2B coding regions (upstream region); from 15 nt downstream of the shift site to 15 nt upstream of the 2B* stop codon (overlap region); and from 30 nt downstream of the 2B* stop codon to 30 nt upstream of the polyprotein stop codon (downstream region). RPF counts were then normalized by the length of the relevant region, and by the total number of counts mapping to all three regions.

To estimate the length of pausing at the mutated shift site we compared the SS and SS-SL mutants of Fig. 2b. In the first round ribosomal profiling, the pausing peak extended over four codons starting at the shift site (Supplementary Fig. 2d) whereas in the second round ribosomal profiling, it was spread over ~20 codons, indicating considerable ribosome run-on during preparation of the latter samples[22]. RPF histograms for SS and SS-SL were first normalized by the total number of RPFs with estimated P-sites mapping within the regions 180 nt after the polyprotein initiation codon to 180 nt upstream of the junction between the 2A and 2B coding regions or 180 nt downstream of the 2B* stop codon to 180 nt upstream of the polyprotein stop codon (Fig. 2c). Then the RPF density was summed over the 20 codons starting from the shift site. Within these 20 codons, SS-SL had a mean normalized RPF density of 0.80 while SS had a mean normalized RPF density of 5.10. Under the assumption that the SS-SL value reflects the effect of local technical biases (PCR, ligation, nuclease), the excess density in SS is $20 \times (5.10/0.80 - 1) = 107.5$, which, assuming a mean translation time of 0.18 s codon$^{-1}$, equates to ~20 s. This calculation is approximate as the underlying RPF distributions over these 20 codons differ between SS (mostly run-on of ribosomes initially paused at the shift site) and SS-SL (codon-specific variability in decoding times) so that an attempt to factor out technical biases via the quotient SS/SS-SL is inherently flawed. Further, it is also possible that pausing may be underestimated if the 3′-adjacent stem–loop and/or bound 2A inhibits nuclease cleavage at the leading edge of the ribosome and the production of appropriately sized RPFs for some fraction of ribosomes paused on the shift site.

**Structure probing.** Short, 33P-labelled RNAs (111 nt) containing the EMCV PRF region (shift site plus 32 nt upstream and 72 nt downstream) were prepared by T7 transcription of a PCR product generated using primers flanking the PRF region, with the 5′ primer containing the T7 polymerase promoter sequence. Transcripts (5 μg) were dephosphorylated with Antarctic phosphatase according to the manufacturer's recommendation (New England BioLabs) and 5′ end-labelled using polynucleotide kinase at 37 °C for 1 h in a reaction containing 20 μCi [γ-33P]ATP (10 mCi ml$^{-1}$ stock; PerkinElmer), 50 mM Tris pH 7.5, 10 mM MgCl$_2$ and 5 mM dithiothreitol. The labelled RNA was loaded onto a 10% acrylamide-urea denaturing gel, full-length RNA eluted from the gel slice in 0.5 M NH$_4$OAc, 10 mM MgOAc, 1 mM ethylenediaminetetraacetic acid (EDTA) and 0.1% SDS and concentrated by ethanol precipitation. RNA structure probing reactions were performed in a final volume of 50 μl containing ~20,000 c.p.m. 5′ [33P] end-labelled transcript, 2 mM MgCl$_2$, 10 μg pig liver rRNA and the relevant enzymatic or chemical probe[40]. Products were analysed on a 10% acrylamide/7 M urea gel.

**RiboTrap and mass spectrometry.** RNA-binding proteins that may associate with the EMCV PRF signal were screened using the RiboTrap method using a commercial kit (RiboCluster Profiler RiboTrap Kit; Medical & Biological

Laboratories, Japan)[41]. WT and scrambled EMCV RNA baits were generated by *in vitro* transcription from plasmid pBKS/EMCV, which contains 126 nt of EMCV sequence beginning immediately 3′-adjacent to the G_GUU_UUU shift site. The 145-nt T7 transcript generated from this vector comprises 126 nt of virus sequence flanked by vector sequences at 5′ (GGGCGAAUUGGAGCU) and 3′ (AAUU) ends. For the scrambled version, the WT stem–loop sequence CG GCA GTG TCA TCA ATG GCT CAA ACC CTA CTG CCG was changed to GA GAA AAC CGC CAC TTC CGA CGC TCG TGT ATG CTC. For generation of Br-U-labelled RNA transcripts, transcription reactions contained a 2:1 ratio of UTP:Br-UTP.

Cell lysates were prepared from 20 T160 flasks of BHK-21 cells infected with EMCV WT at MOI ∼10, or mock infected. Cells were scraped into the medium, collected by centrifugation at 300*g* for 5 min at 4 °C and washed three times with ice-cold PBS before resuspension in 2,400 μl of RiboTrap CE buffer supplemented with DTT to a final concentration of 1.5 mM. The lysate was incubated on ice for 10 min, 120 μl of RiboTrap Detergent solution added and the tubes inverted gently five times. The lysate was centrifuged immediately for 5 min at 3,000*g* to precipitate the nuclei. Seventy-two microlitres of RiboTrap High-Salt solution was added to the tube, mixed gently and centrifuged at 12,000*g* for 3 min at 4 °C. The supernatant was retained and used immediately in the RiboTrap assay. RNA-binding proteins associated with the target RNA were analysed by SDS–PAGE on a 4–20% tris-glycine gel. Coomassie blue-stained products were excised from the gel and subjected to in-gel trypsin digestion. Peptides were extracted and analysed by liquid chromatography tandem mass spectrometry (LC-MS/MS). Fragment MS/MS spectra were searched with the MASCOT search engine (Matrix Science) against a protein sequence database composed of expected viral target sequences, cellular proteins and common contaminant proteins.

**Protein expression and purification.** N-terminally glutathione-*S*-transferase-tagged proteins were purified from *E. coli* BL21/DE3/pLysS cells. Single colonies were picked into Luria-Bertani broth and grown at 37 °C to an $OD_{600}$ of 0.6. Protein expression was induced by addition of isopropyl β-D-1-thiogalactopyranoside (to 0.1 mM) and continued for 2 h at 37 °C (or overnight at 22 °C) after which cells were pelleted and resuspended in lysis buffer (0.02 M β-mercaptoethanol, 0.5 mM $MgCl_2$, 0.05% Tween 20, 0.02 M Tris pH 7.5, 150 mM NaCl, DNase 1 U ml$^{-1}$ and protease inhibitor 1 U ml$^{-1}$). Cells with protease inhibitor cocktail (Sigma) and DNase (Sigma) were incubated on ice for 30 min and sonicated to complete lysis. Proteins were purified using glutathione agarose resin (GE Healthcare) according to the standard procedures[42], then dialysed against 50 mM Tris, pH 7.5, 100 mM KCl, 1 mM dithiothreitol (DTT), 0.05 mM EDTA and 5% glycerol, quantified by Bradford assay (ThermoFisher Scientific), and stored at −80 °C until required.

**Immunoblotting.** Infected cell lysates or purified proteins were separated on 15% acrylamide/bisacrylamide gels (BioRad mini-protean tetra cell apparatus) and transferred to nitrocellulose membranes for 120 min using a mini-protean transblot cell and Tris glycine transfer buffer. Following blocking with 5% non-fat milk, primary antibody incubations were carried out overnight at 4 °C using 1:1,000 diluted anti-2A (rabbit polyclonal raised against the C-terminal 14 aa of 2A by GenScript) or anti-tubulin (rat monoclonal; Abcam, ab6160) antibodies. Secondary antibody was the appropriate IRDye-conjugate (Li-Cor) used at 1:10,000 dilution. All washes employed PBS containing 0.1% Tween 20. Blots were scanned on a Licor Odyssey infrared scanner.

**Electrophoretic mobility shift assay.** Short, $^{32}$P-labelled template RNAs (64 nt) containing the EMCV PRF region (with slippery sequence precisely at the 5′ end) were prepared by T7 transcription of a PCR product generated using primers flanking the PRF region, with the 5′ primer containing the T7 polymerase promoter sequence. Radiolabelled RNAs were mixed with test proteins in 10 μl reactions in EMSA buffer (10 mM Hepes pH 7.6, 150 mM KCl, 2 mM $MgCl_2$, 1 mM DTT, 0.5 mM adenosine triphosphate, 5% glycerol, 100 μg ml$^{-1}$ porcine tRNA, 10 U RNase inhibitor ml$^{-1}$). Test proteins were diluted in dilution buffer (DB) (5 mM Tris pH 7.5, 100 mM KCl, 1 mM DTT, 0.05 mM EDTA, 5% glycerol). For competition experiments, unlabelled competitor RNA was incubated with WT $^{32}$P-labelled RNA and 2A (1.8 μM). Reactions were incubated at 30 °C for 10 min before promptly loading the mix onto 4% acrylamide non-denaturing gels (acrylamide:bisacrylamide ratio 10:1). Gels were run at 175 V at room temperature until free and bound RNA species were resolved, then fixed for 15 min in 10% acetic acid, 10% methanol, dried and exposed to X-ray film and phosphor-imager screen.

**In vitro translation.** Frameshift reporter plasmids were linearized with *Fsp*I and capped run-off transcripts generated using T7 RNA polymerase[43]. Messenger RNAs were translated in nuclease-treated RRL or WG extracts (Promega) programmed with ∼50 μg ml$^{-1}$ template mRNA. Typical reactions were of 10 μl volume and composed of 90% (v/v) RRL, 20 μM amino acids (lacking methionine) and 0.2 MBq [$^{35}$S]-methionine. Reactions were incubated for 1 h at 30 °C and stopped by the addition of an equal volume of 10 mM EDTA, 100 μg ml$^{-1}$ RNase A followed by incubation at room temperature for 20 min.

Proteins were resolved by 12% SDS–PAGE and dried gels were exposed to X-ray film or to a Cyclone Plus Storage Phosphor Screen (PerkinElmer). The screen was scanned using a Typhoon TRIO Variable Mode Imager (GE Healthcare) in storage phosphor autoradiography mode and bands were quantified using ImageQuantTL software (GE Healthcare). PRF efficiencies were calculated as [IFS/MetFS]/[IS/MetS + IFS/MetFS], where the number of methionines in the stop and frameshift products are denoted by MetS and MetFS, respectively, and the densitometry values for the same products are denoted by IS and IFS, respectively. All frameshifting assays were performed at least three times.

**Ribosome pausing assays.** WG and RRL *in vitro* translation reactions (30 μl) were supplemented with 1 μM of 2A, 2A-mut or dialysis buffer, and programmed with mRNAs derived from *Ava*II-cut pPS-EMCV-WT or mutant derivatives. Reactions were incubated at 26 °C for 5 min prior to the addition of edeine to 5 μM final concentration. Aliquots (1.5 μl) were subsequently withdrawn at set intervals post-edeine addition, mixed with an equal volume of 100 μg ml$^{-1}$ RNase A in 10 mM EDTA and placed on ice. At the end of the time-course, products were resolved by 12% SDS–PAGE. The expected size of the ribosomal pause product was marked by translating a control mRNA produced from *Xho*I-cleaved pPS0.

**Data availability.** Sequencing data have been deposited in ArrayExpress (http://www.ebi.ac.uk/arrayexpress) under the accession number E-MTAB-5206. The data that support the findings of this study are available from the corresponding author upon request.

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

## Acknowledgements

This work was supported by Wellcome Trust grants (088789) and (106207), UK Biotechnology and Biological Research Council (BBSRC) grant (BB/J007072/1) and a European Research Council (ERC) European Union's Horizon 2020 research and innovation programme grant (646891) to A.E.F.; by BBSRC grant (BB/L000334/1) and UK Medical Research Council grant (MR/M011747/1) to I.B.; by a BBSRC PhD studentship to L.K.F.; and by a Wellcome Trust PhD scholarship to J.D.J. We thank Gary Loughran, John Atkins, Ann Palmenberg, Paul Digard, Trevor Sweeney, Steven Graham and Ian Goodfellow for materials and helpful discussions.

## Author contributions

A.E.F. and I.B. conceived and supervised the project. I.B. and J.D.J. performed the ribosome profiling. A.E.F. and J.D.J. analysed the profiling data. L.K.F. performed RiboTrap and initial *in vitro* analyses. S.N. performed the structure probing, EMSA, and *in vitro* frameshifting and pausing analyses. Generation and characterization of new virus mutants were performed by S.N. and S.B. (2A-mut) and R.L. (all other mutants). S.B. prepared recombinant proteins and provided technical support. A.E.F. wrote the manuscript. All authors edited the manuscript.

## Additional information

**Competing interests:** The authors declare no competing financial interests.

