## [Peer Review File · Nature Communications]

Reviewers' Comments:

Reviewer #1 (Remarks to the Author):

In this manuscript the authors described a novel role of EMCV 2A protein in Programmed -1 ribosomal frameshifting. The authors first used ribosome profiling to quantitatively measure the frameshifting efficiency over the course of EMCV infection. Then they combined ribosome profiling and biochemical characterizations and showed that 2A binds to an RNA stem-loop downstream of the slippery sequence that induces highly efficient PRF. The increase of frameshifting efficiency at late time of infection likely serves as a control mechanism to regulate viral structural and non-structural protein synthesis during the replicative cycle. The novelty lies in the identification of a paradigm of temporal control of viral gene expression and elucidation of a previously uncharacterized -1 PRF mechanism. The data are interesting and the discovery is important. Addressing the following points will strengthen the evidence to support the author's conclusions.

Major points:

1. The ribosome profiling sequencing quality was not adequately described in the manuscript. High sequencing data quality and reproducibility would improve the confidence of the analyses and conclusions. Especially, the percentages of mapped viral and host reads in total reads (Table S1) are quite low. Please explain. The data quality can be assessed by one or more ways found in the following papers:

Ingolia et al (2011), Cell 147, 789-802.

Tirosh et al (2015), PLoS Pathog. 11(11): e1005288.

Dai et al (2016), J Virol. pii: JVI.01858-16. doi: 10.1128/JVI.01858-16.

2. Because the kinetics of RNA synthesis are not synchronous with the kinetics of RNA translation, and the newly synthesized RNA may be destined for packaging rather than translation, the authors justified that RNA-Seq densities was not used to normalize ribosome profiling reads in analysis. However, such analyses are flawed due to the omission of potential difference of RNA levels on different regions of viral genome, which could contribute to the difference of the calculated frameshifting efficiency. Are there sub-genomic RNA evidenced in EMCV infection? At least a parallel analysis using RNA-seq normalized reads is necessary to rule out this concern.

3. Related to the 2nd point, while the authors provided parallel RNA-seq map at late infection for 8hpi (Fig. S3), there is no parallel RNA-seq data presented for 2, 4, and 6 hpi for Fig. 1D and Fig. S1A. Providing such data would also reduce the concern of RNA contribution to the calculated frameshifting efficiency difference.
4. While the authors thought that the difference in frameshifting efficiency could be correlated with the expression of 2A, a Western blot of 2A protein levels corresponding to the ribosome profiling time points would make it more evident.
5. The authors used in vitro translation assays to show the role of 2A in the frameshifting translation. An in vivo experiment to strength the conclusion could be by exogenous expressing 2A prior EMCV infection and examine its effect on frameshifting and EMCV replication.
6. I am curious whether similar stem loops for frameshifting are present in other piconaviruses, such as poliovirus, EV71, etc. Such a bioinformatics investigation would give a clue whether this frameshifting mechanism is also used by other piconaviruses.

Minor points

1. Though the authors have cited references for ribosome profiling and RNA-Seq, a brief description of the procedure is still necessary.
2. Did the authors only analyzed ribosome profiling reads around 28-30 nts? Which is not described in the manuscript.
3. Are the SL mutations in Fig. 2A silent mutations? Do they change amino acid sequence? If so, could they affect the frameshifting?
4. It is not clear why the SL stem-loop mutant wasn't used in the 2A binding assay in Fig. 3.
5. Fig. 5, what would the result be when use WT RNA+2A-mut?
6. Line 428, RNase 1 to RNase I.

Reviewer #2 (Remarks to the Author):

The authors demonstrate convincingly that the EMCV-encoded 2A protein stimulates -1 programmed ribosomal frameshifting at a site immediately downstream of the 2A region, within the region encoding the 2B protein. The abundance of 2A during infection increases with time

and thus the frequency of frameshifting increases from an initial very low level, undetectable in these studies, to as high as 70%. They demonstrate this with an impressive set of experiments exploiting ribosome profiling, detailed mutagenesis and ultimately showing that in a heterologous system in the absence of EMCV infection the 2A protein stimulates frameshifting and pausing at the site of the frameshift (especially on RNAs in which the frameshift heptamer has been disabled by mutation). For mutants in which frameshifting is disabled, the protein induces a strong and persistent pause at the mutant frameshift site. This is true where the heptamer is mutated but also when a downstream RNA hairpin is mutated including at a site required for the 2A protein to bind it, and where a positively-charged region of 2A required for hairpin binding is mutated. These studies comprehensively demonstrate that pausing and frameshifting depend on binding of 2A protein to the downstream hairpin at a CCC region in the hairpin loop. The fact that the protein also induces a strong translational pause where it is impossible for the ribosome to frameshift suggests that pausing might be a prerequisite for frameshifting, the authors argue, and not just a side reaction, as has been suggested by them and others.

The authors had previously shown that the combination of a viral and cellular protein can stimulate -2 programmed ribosomal frameshifting in members of the family Arterioviridae. In that case they had not shown the temporal effect seen in EMCV. Others have shown that a small RNA can stimulate -1 frameshifting but again without the temporal effect. The fact that the spacing between the frameshift heptamer and the downstream stem loop is much larger than in canonical -1 simultaneous slippage frameshifting suggests that attempts to identify frameshift sites informatically, which depended on a consistent smaller spacing, may have missed other examples of protein-induced frameshifting. For that reason, it is possible that this mechanism is more common than previously suspected in chromosomal genes.

I have only one critical comment and this is technical. In the electrophoretic mobility shift assays the authors express the protein concentration in the figures in terms of fold molar excess and they do not report (directly) the concentration of the radioactively labeled RNA probe. Since these assays depend on the probe being present in sufficiently low concentration relative to the protein to avoid a driving effect of the RNA on formation of the protein-RNA hybrid, it is essential that it be present in much lower concentration than the protein. From the data presented in Figure 3B it appears that the RNA is present at about 10nM concentration and the protein in from 450 nM to 12 μ M with an apparent K_d of 3-4.5 μ M. The fold molar excess is actually determined by the concentration of the RNA probe, so that if it were say 10-fold less concentrated the fold molar excess would be increased 10-fold, but the concentration of protein giving 50% binding would have been unchanged at 3-4.5 μ M. For these reasons, the figure should be labeled entirely in molarity and the molar concentration of the RNA should be given in the text.

Reviewer #3 (Remarks to the Author):

In this manuscript the authors demonstrate that during EMCV infection a -1 frameshift occurs in temporally regulated manner. Early in infection the ribosomes translate the full polyprotein whereas late in infection up to 70% of the ribosomes are frame-shifting. This regulated frameshift changes the relative expression of structural and enzymatic proteins during infection. The authors further demonstrate that the frame-shift is trans-activated by the viral protein 2A, suggesting a novel control mechanism in which late in infection (as a result of increasing concentrations of 2A) frame shifting becomes highly efficient facilitating this temporal regulation.

The work is of great interest as it illustrates for the first time that frame-shifting is used to temporally regulate gene expression. Overall the manuscript is well written; the data is convincing and supports the authors conclusions.

Few comments that are aimed to improve some aspects in the manuscript:

The model the authors provide is that frameshifting acts as a mechanism to free ribosomes by acting to reduce the number of ribosomes translating the 3'-encoded enzymatic proteins, freeing more ribosomes to translate the 5'-encoded structural proteins. Although this model is appealing, it suggests that ribosomes are a major limiting factor during infection which is not necessarily the case. I wonder whether the authors can test this hypothesis by calculating whether the % of ribosome footprints that originate from the virus stay constant between the WT and frame shift mutants (relative to the human) and only the main change is in the distribution of reads.

The authors should keep in mind there could be alternative explanations for the necessity of the frame-shifting; a functional role of 2B*, toxicity from over producing enzymatic protein etc.

To better understand the functional importance of the frame-shift, it will help to better describe the defects that were observed in the frame-shift mutants. Was the phenotype of the various mutants (in which the frameshifting is abolished) similar? Was the LV-SS-SL defective compared to LV-WT?

In figures 1 and 2 the authors present the frame information they obtain from the ribosome protected fragments. Surprisingly, RPF 5' ends map mainly to the 1st and 3rd positions of polyprotein codons. Why is there also a strong preference to frame 3? Is this the result of an incomplete digestion of the sample? To reassure it is not a phenomenon that is related to the virus it will be good to present the same data also for host mRNAs from the same samples.

In figure 2C the authors calculate the excess accumulation of RPFs in SS mutant and calculate pause of ~20 s. However, in different panels in the manuscript the height of the pause looks

variable (figure 1A, Figure S 1 and figure 2C) unless this calculation is based on several samples, or that the visual inspection is misleading I think the authors should be more careful in calculating the length of the pause based on this data.

We thank all three reviewers for their positive assessment of the manuscript and helpful suggestions for improvements. A detailed point-by-point response follows below.

Reviewers' comments:

Reviewer #1 (Remarks to the Author):

In this manuscript the authors described a novel role of EMCV 2A protein in Programmed -1 ribosomal frameshifting. The authors first used ribosome profiling to quantitatively measure the frameshifting efficiency over the course of EMCV infection. Then they combined ribosome profiling and biochemical characterizations and showed that 2A binds to an RNA stem-loop downstream of the slippery sequence that induces highly efficient PRF. The increase of frameshifting efficiency at late time of infection likely serves as a control mechanism to regulate viral structural and non-structural protein synthesis during the replicative cycle. The novelty lies in the identification of a paradigm of temporal control of viral gene expression and elucidation of a previously uncharacterized -1 PRF mechanism. The data are interesting and the discovery is important. Addressing the following points will strengthen the evidence to support the author's conclusions.

Major points:

1. The ribosome profiling sequencing quality was not adequately described in the manuscript. High sequencing data quality and reproducibility would improve the confidence of the analyses and conclusions. Especially, the percentages of mapped viral and host reads in total reads (Table S1) are quite low. Please explain. The data quality can be assessed by one or more ways found in the following papers:

Ingolia et al (2011), Cell 147, 789-802.

Tirosh et al (2015), PLoS Pathog. 11(11): e1005288.

Dai et al (2016), J Virol.. pii: JVI.01858-16. doi: 10.1128/JVI.01858-16.

The percentages of mapped viral RNA and host mRNA reads in Table S1 reflect the fact that we didn't use any rRNA depletion protocol, such as RiboZero or other oligo-based depletion strategies or double-stranded nuclease (DSN) (Chung et al, 2015, *RNA*, PMID 26286745). We did this because the viral RNA is present in high amounts compared to individual host mRNA species and therefore improving yield via rRNA depletion was unnecessary. This in no way affects the quality of the data and indeed, avoids any possible biases that such methods might introduce. For virus RNA, our depth of coverage is more than sufficient for our analyses. To clarify this point, we have added a column to Table S1 showing the number of reads mapping to rRNA and added to the Methods section the phrase "RiboZero-based rRNA subtraction was used for RNA-Seq libraries while Ribo-Seq libraries were untreated."

We always perform a range of quality checks on our virus ribosome profiling datasets. Indeed, in a previous publication on murine coronavirus, we went into great detail on bioinformatic methods to assess the quality of virus profiling datasets, and discussion of potential contamination problems (Irigoyen et al., 2016, *PLoS Path*, PMID 26919232). We felt that such extensive detail was perhaps better omitted here given the concise format of Nature Communications and the focus of the current work. Further, some host mRNA analyses are of limited relevance to this manuscript given that the manuscript is only concerned with virus translation, and at late timepoints host mRNA coverage is extremely low due to virus-induced shut-off of host mRNA translation.

Nonetheless, we have now added host mRNA phasing, read length distributions, and summed-over-host-mRNA RPF mean density plots to the supplementary material (now Figs S1 and S4).

2. Because the kinetics of RNA synthesis are not synchronous with the kinetics of RNA translation, and the newly synthesized RNA may be destined for packaging rather than translation, the authors justified that RNA-Seq densities were not used to normalize ribosome profiling reads in analysis. However, such analyses are flawed due to the omission of potential differences in RNA levels on different regions of the viral genome, which could contribute to the difference in the calculated frameshifting efficiency. Are there sub-genomic RNAs evidenced in EMCV infection? At least a parallel analysis using RNA-seq normalized reads is necessary to rule out this concern.

As detailed in the manuscript, we normalize the Ribo-Seq profiles of WT and mutant viruses by the SS mutant, which provides extremely good normalization for the question we are addressing. RNA-Seq densities would only have an effect if RNA-Seq profiles differed between WT or mutants and SS. However, Figure S5 (RNA-Seq profiles) makes it very clear that the massive drop-off in ribosome density seen only in WT and LV-WT-SL (Figures 1D and 2B) cannot be explained by changes in RNA-Seq.

Like other members of the family *Picornaviridae*, EMCV does not produce subgenomic RNAs. Indeed, it would be hard to explain a 5' to 3' ribosome density gradient on the basis of sgRNA as such 3'-truncated transcripts would lack stop codons, leading to ribosome stalling at their 3' ends and initiation of "non-stop" mRNA and nascent protein decay mechanisms.

In early work on this project, we did investigate the suggested parallel analysis using RNA-Seq normalization. The downstream/upstream ratios calculated in this way agree to within 6% (relative difference) of the values calculated using the methodology given in the manuscript for WT, WT-SL, SS-SL, LV-WT and LV-SS-SL at 8 h p.i. Thus there is actually very little difference between our method and the RNA-Seq normalized method.

We believe that the analysis reported in the manuscript (normalizing WT Ribo-Seq by shift site mutant Ribo-Seq) is the correct way to do the analysis. We'd be happy to add the RNA-Seq normalized results into the manuscript but we prefer not to as we believe it isn't the right way to do the analysis and thus distracts from the key messages.

3. Related to the 2nd point, while the authors provided a parallel RNA-seq map at late infection for 8 hpi (Fig. S3), there is no parallel RNA-seq data presented for 2, 4, and 6 hpi for Fig. 1D and Fig. S1A. Providing such data would also reduce the concern of RNA contribution to the calculated frameshifting efficiency difference.

We performed RNA-Seq for (most of) the time-course as well but excluded it from the manuscript because, as detailed above, it was not relevant to the analysis. Again, there was no difference in the conclusions when RNA-Seq normalized values were used in the calculations.

4. While the authors thought that the difference in frameshifting efficiency could be correlated with the expression of 2A, a Western blot of 2A protein levels corresponding to the ribosome profiling time points would make it more evident.

We have now added a Western blot time-course of 2A levels in infected cells (now Figure 4D) showing that the amount of 2A increases dramatically over the course of infection and its appearance correlates with the stimulation in frameshifting efficiency seen at 6 h p.i. and beyond (Figure 1E).

5. The authors used in vitro translation assays to show the role of 2A in the frameshifting translation. An in vivo experiment to strengthen the conclusion could be by exogenously expressing 2A prior to EMCV infection and examining its effect on frameshifting and EMCV replication.

This is indeed an interesting experiment on which we are working. A problem we face is that 2A has a toxic effect on transfected cells (perhaps through its interference with cap-dependent translation), and we have as yet been unable to tease apart the potential detrimental effects of premature efficient frameshifting from the detrimental effects of reduced cell viability on virus replication. The 2A-mutant virus in the current manuscript (now Figures 4A, B and C) however already provides direct *in vivo* (cell culture) evidence of the role of 2A in frameshifting.

6. I am curious whether similar stem loops for frameshifting are present in other piconaviruses, such as poliovirus, EV71, etc. Such a bioinformatics investigation would give a clue whether this frameshifting mechanism is also used by other piconaviruses.

Although many piconaviruses have a 2A protein, the origin and function of the protein encoded at this genomic location differs between different picornavirus genera. Notably, the cardiovirus 2A protein is not homologous to other picornavirus 2As, and thus the identification of (protein-stimulated) frameshifting in other piconaviruses is not simply a matter of looking for other CCC-containing stem-loops in a central region of the genome. There are various approaches that might be used to find other examples, including (i) comparative genomics (we have identified signals in some other picorna-like viruses that are potential candidates but require full experimental characterization before making any such claim in print), and (ii) ribosome profiling (beyond the scope of the current manuscript but planned for future work).

Minor points

1. Though the authors have cited references for ribosome profiling and RNA-Seq, a brief description of the procedure is still necessary.

We already have a schematic of the ribosome profiling procedure (Figure 1B) and the following text in the figure caption, "Schematic of the ribosome profiling strategy. Each translating ribosome protects ~30 nt of mRNA. Cells are lysed, treated with RNase I to degrade unprotected mRNA, ribosomes are harvested, and the ribosome protected fragments (RPFs) extracted and subjected to high-throughput sequencing."

2. Did the authors only analyzed ribosome profiling reads around 28-30 nts? Which is not described in the manuscript.

We made no selection on the length of ribosome profiling reads used in the analysis. As expected, our length distributions are typically tightly peaked around 30 nt, so we saw no reason to potentially bias the results by making such restrictions *in silico*.

3. Are the SL mutations in Fig. 2A silent mutations? Do they change amino acid sequence? If so, could they affect the frameshifting?

The SL mutations are silent in the polyprotein reading frame (Materials and Methods, last sentence of subsection "Cells, recombinant viruses and plasmids").

4. It is not clear why the SL stem-loop mutant wasn't used in the 2A binding assay in Fig. 3.

The SL mutant had three synonymous point mutations spread over a 10-nt region for ribosome profiling analysis of virus-infected cells. For the *in vitro* assays, we wanted to dissect the role of individual point mutations. Since the single mutation C46U inhibited binding, testing all three mutations together would not have added anything.

5. Fig. 5, what would the result be when use WT RNA+2A-mut?

We have now added this figure to the manuscript (now Figure S8D). Since WT RNA has much less pausing than SS RNA in the presence of 2A, and since SS + 2A-mut has no pausing, not surprisingly the WT RNA + 2A-mut protein also shows no pausing (and no frameshift product).

6. Line 428, RNase 1 to RNase I.

This error has been corrected.

Reviewer #2 (Remarks to the Author):

The authors demonstrate convincingly that the EMCV-encoded 2A protein stimulates -1 programmed ribosomal frameshifting at a site immediately downstream of the 2A region, within the region encoding the 2B protein. The abundance of 2A during infection increases with time and thus the frequency of frameshifting increases from an initial very low level, undetectable in these studies, to as high as 70%. They demonstrate this with an impressive set of experiments exploiting ribosome profiling, detailed mutagenesis and ultimately showing that in a heterologous system in the absence of EMCV infection the 2A protein stimulates frameshifting and pausing at the site of the frameshift (especially on RNAs in which the frameshift heptamer has been disabled by mutation). For mutants in which frameshifting is disabled, the protein induces a strong and persistent pause at the mutant frameshift site. This is true where the heptamer is mutated but also when a downstream RNA hairpin is mutated including at a site required for the 2A protein to bind it, and where a positively-charged region of 2A required for hairpin binding is mutated. These studies comprehensively demonstrate that pausing and frameshifting depend on binding of 2A protein to the downstream hairpin at a CCC region in the hairpin loop. The fact that the protein also induces a strong translational pause where it is impossible for the ribosome to frameshift suggests that pausing might be a prerequisite for frameshifting, the authors argue, and not just a side reaction, as has been suggested by them and others.

The authors had previously shown that the combination of a viral and cellular protein can stimulate -2 programmed ribosomal frameshifting in members of the family Arterioviridae. In that case they had not shown the temporal effect seen in EMCV. Others have shown that a small RNA can stimulate -1 frameshifting but again without the temporal effect. The fact that the spacing between the frameshift heptamer and the downstream stem loop is much larger than in canonical -1 simultaneous slippage frameshifting suggests that attempts to identify frameshift sites informatically, which depended on a consistent smaller spacing, may have missed other examples of protein-induced frameshifting. For that reason, it is possible that this mechanism is more common than previously suspected in chromosomal genes.

I have only one critical comment and this is technical. In the electrophoretic mobility shift assays the authors express the protein concentration in the figures in terms of fold molar excess and they do not report (directly) the concentration of the radioactively labeled RNA probe. Since these assays depend on the probe being present in sufficiently low concentration relative to the protein to avoid a driving effect of the RNA on formation of the protein-RNA hybrid, it is essential that it be present in much lower concentration than the protein. From the data presented in Figure 3B it appears that the RNA is present at about 10nM concentration and the protein in from 450 nM to 12 μ M with an apparent K_d of 3-4.5 μ M. The fold molar excess is actually determined by the concentration of the RNA probe, so that if it were say 10-fold less concentrated the fold molar excess would be increased 10-fold, but the concentration of protein giving 50% binding would have been unchanged at 3-4.5 μ M. For these reasons, the figure should be labeled entirely in molarity and the molar concentration of the RNA should be given in the text.

While we appreciate the intent of labelling the figure entirely in molarity, we note that it is also quite customary to label EMSA figures in molar excess provided the RNA molarity is given in the figure caption, as it is in the caption to Figure 3. Overall, we prefer this presentation style and all of the information is still there.

Reviewer #3 (Remarks to the Author):

In this manuscript the authors demonstrate that during EMCV infection a -1 frameshift occurs in temporally regulated manner. Early in infection the ribosomes translate the full polyprotein whereas late in infection up to 70% of the ribosomes are frame-shifting. This regulated frameshift changes the relative expression of structural and enzymatic proteins during infection. The authors further demonstrate that the frame-shift is trans-activated by the viral protein 2A, suggesting a novel control mechanism in which late in infection (as a result of increasing concentrations of 2A) frame shifting becomes highly efficient facilitating this temporal regulation.

The work is of great interest as it illustrates for the first time that frame-shifting is used to temporally regulate gene expression. Overall the manuscript is well written; the data is convincing and supports the authors conclusions.

Few comments that are aimed to improve some aspects in the manuscript:

The model the authors provide is that frameshifting acts as a mechanism to free ribosomes by acting to reduce the number of ribosomes translating the 3'-encoded enzymatic proteins, freeing more ribosomes to translate the 5'-encoded structural proteins. Although this model is appealing, it suggests that ribosomes are a major limiting factor during infection which is not necessarily the case. I wonder whether the authors can test this hypothesis by calculating whether the % of ribosome footprints that originate from the virus stay constant between the WT and frame shift mutants (relative to the human) and only the main change is in the distribution of reads.

The authors should keep in mind there could be alternative explanations for the necessity of the frame-shifting; a functional role of 2B*, toxicity from over producing enzymatic protein etc.

To better understand the functional importance of the frame-shift, it will help to better describe the defects that were observed in the frame-shift mutants. Was the phenotype of the various mutants (in which the frameshifting is abolished) similar? Was the LV-SS-SL defective compared to LV-WT?

[Addressing the combined comments above:]

We agree that frameshifting as a mechanism to liberate more ribosomes for structural protein synthesis is probably only part of the story and consequently we only mentioned it briefly in the manuscript (Discussion "providing a translational bias that favours virion protein production at late timepoints"). Even if ribosomes are limiting, frameshifting would only increase structural protein synthesis by ~2-fold. Instead, our thinking is that it is more to do with temporal control of the expression levels of structural and non-structural protein synthesis (lines 26, 78-83, 248-249), potentially as part of the switch from virus replication to virion production, and also, as the reviewer rightly points out, to avoid potential toxicity from over producing enzymatic protein.

Regarding the suggestion to nonetheless test the possibility that frameshifting is a mechanism to liberate more ribosomes for structural protein synthesis by calculating whether the % of ribosome

footprints that originate from the virus stay constant between the WT and frame shift mutants (relative to the host) and only the main change is in the distribution of reads: Since frameshifting removes 70% of the ribosomes from 49% of the polyprotein ORF in WT at 8 h p.i., we'd expect an ~34% reduction in virus RPFs in WT relative to SS if ribosomes are not limiting. We don't see evidence for such large differences (e.g. in our first round of ribosome profiling, WT virus accounts for 67% of total cellular translation at 8 h p.i. whereas SS virus accounts for 56%). However, an accurate analysis with our current data sets is not possible due to sample-to-sample variations (which could be due to various reasons, but at least would require more accurate characterization of WT and mutant virus growth characteristics).

That 2B* also plays a functional role in itself is almost certainly correct. In previous work (Loughran et al., 2011, *PNAS*), synonymous-site conservation in the polyprotein frame was observed throughout the 117-codon overlapping 2B* ORF, considerably 3' of the signals now shown to be sufficient for frameshifting, indicating that 2B* itself is subject to purifying selection. However in that paper we also showed that shift site mutants were more attenuated than 2B*-truncation mutants (all synonymous in the polyprotein frame) indicating that frameshifting *per se* is also important. Of course in this previous work we had no idea about the role of 2A in frameshifting, the frameshifting mechanism, or the temporal regulation.

The current manuscript includes phenotyping data for the SS and 2A-mut viruses compared to WT (now Figure 4A,B,C). The other mutants are also attenuated to varying extents but are not fully characterised yet in this regard. Further, we do not feel that this mechanistically oriented manuscript is the best place for virus phenotyping. An accurate disentanglement of 2B* effects from frameshifting *per se* might be possible via characterization of WT and SS viruses in 2B*-expressing cells but there are many confounding factors (e.g. expression level and localization issues) and, again, this more virologically focused work would be more appropriately placed in a separate manuscript. Interestingly, in the related Theiler's murine encephalomyelitis virus, the overlapping ORF has just 8 codons leading to a 2B* protein of just 14 amino acids (Figure S3A).

In figures 1 and 2 the authors present the frame information they obtain from the ribosome protected fragments. Surprisingly, RPF 5' ends map mainly to the 1st and 3rd positions of polyprotein codons. Why is there also a strong preference to frame 3? Is this the result of an incomplete digestion of the sample? To reassure it is not a phenomenon that is related to the virus it will be good to present the same data also for host mRNAs from the same samples.

We have added host mRNA RPF phasing plots for the different samples (now Figures S1C and S4C). They are very similar to the virus phasing plots or indeed have worse phasing in some cases (this may be attributable, potentially, to contaminants such as RNP footprints which will be more apparent than in other studies against the background of extremely low host translation due to virus-induced shut off of cap-dependent translation). As viral translation is substantially higher than the sum total of all host mRNA translation in these samples, contamination is much less of an issue in the virus data.

The reviewer is correct that the high-low-high phasing pattern is most likely a result of incomplete digestion - RPFs mapping to the 3rd codon position generally have 1 additional nucleotide at the 5' end of the read. It is a curious situation that two Ribo-Seq practioners in our group consistently obtain this high-low-high pattern while the other two consistently obtain a high-low-low pattern. Nonetheless, the fact that the EMCV data has a high-low-high pattern does not in any way affect our analyses and still allows us to quality assess our data. The reviewer's point is a good one though - we also have wondered about the possibility of modified ribosomes, potentially with a non-canonical footprint size, on virus mRNAs, although there is no evidence for that in this case.

In figure 2C the authors calculate the excess accumulation of RPFs in SS mutant and calculate pause of ~20 s. However, in different panels in the manuscript the height of the pause looks

variable (figure 1A, Figure S 1 and figure 2C) unless this calculation is based on several samples, or that the visual inspection is misleading I think the authors should be more careful in calculating the length of the pause based on this data.

This is an astute observation; there are a couple of reasons for the variation, and as a result of them we had to take care in the way we made the pausing calculation. The variation between Figures 1 and S1 (now Figure S2) is due to the magnitude of the pause increasing over the course of infection. This is readily apparent in the single-nucleotide resolution plot in what are now Figures S2C and D. The reason for the difference between Figure 1D and Figure 2B/C (biological repeats of SS at 8 h p.i.) is due to increased ribosome run-on - i.e. slower arrest from cycloheximide treatment - in the latter sample (Methods lines 436-439). Thus the ribosomes initially paused at the shift site spread out over a larger number of codons during sample preparation in the latter than in the former, leading to a reduction in the peak height but a similar total excess density over the base level density. The reasons for these differences in sample preparation are unknown. The pausing calculation was only performed for the round 2 profiling since only the SS versus SS-SL comparison is properly controlled. Even then, the method comes with some caveats (Methods lines 447-453).

Nonetheless, we feel that this is an important calculation to include in the paper: to our knowledge, this is the first estimate of the pausing time associated with ribosome frameshift stimulators that comes directly from translation in cells as opposed to *in vitro* translation systems (some of which are highly artificial). (Incidentally, the estimate of ~20 s is almost identical - possibly coincidentally - to the pausing time for *E. coli* ribosomes paused on a mutated shift site with canonical 3' RNA pseudoknot frameshifting stimulator measured in a cell-free translation system using the stopped-flow technique; Caliskan et al., 2014, *Cell*.) While there are caveats associated with our calculation, it gives a good idea that it is much more than a mean translocation time (0.18^{-1} s) and much less than 10 mins (approximate time to translate the virus genome).

Although we have already outlined the caveats in the Methods, we have now modified the main text from "equates to a ribosomal pause of ~20 s." to "equates to a ribosomal pause of ~20 s (albeit with certain caveats; see Methods)."

Reviewers' Comments:

Reviewer #1 (Remarks to the Author):

The revision addressed most of my concerns. I have no further questions.

Reviewer #3 (Remarks to the Author):

I have no additional comments